# Target Therapy for Extramedullary Relapse of *FLT3*-ITD Acute Myeloid Leukemia: Emerging Data from the Field

**DOI:** 10.3390/cancers14092186

**Published:** 2022-04-27

**Authors:** Andrea Duminuco, Cinzia Maugeri, Marina Parisi, Elisa Mauro, Paolo Fabio Fiumara, Valentina Randazzo, Domenico Salemi, Cecilia Agueli, Giuseppe Alberto Palumbo, Alessandra Santoro, Francesco Di Raimondo, Calogero Vetro

**Affiliations:** 1Postgraduate School of Hematology, University of Catania, 95123 Catania, Italy; andrea.duminuco@gmail.com (A.D.); giuseppe.palumbo@unict.it (G.A.P.); 2Division of Hematology, A.O.U. “Policlinico G.Rodolico-S.Marco”, 95123 Catania, Italy; maugericinzia@hotmail.com (C.M.); marinaparisi@hotmail.it (M.P.); elixmauro@hotmail.it (E.M.); paolo.fiumara@virgilio.it (P.F.F.); diraimon@unict.it (F.D.R.); 3Division of Hematology & Bone Marrow Transplantation, Ospedali Riuniti Villa Sofia-Cervello, 90146 Palermo, Italy; valentina.randazzo@hotmail.it (V.R.); d_salemi@gmail.it (D.S.); c.agueli71@gmail.com (C.A.); a.santoro@villasofia.it (A.S.); 4Department of Biomedical and Biotechnological Sciences, University of Catania, 95123 Catania, Italy; 5Department of Chirurgia Generale e Specialità Medico-Chirurgiche, University of Catania, 95123 Catania, Italy

**Keywords:** myeloid sarcoma, extramedullary manifestation of AML, *FLT3*, acute myeloid leukemia

## Abstract

**Simple Summary:**

Extramedullary presentation of acute myeloid leukemia is a rare event that can occur at diagnosis or at relapse, together or independent of bone marrow involvement. This event is usually unamenable to standard therapies. Due to the rarity of this disorder, most of the literature comprises small retrospective studies and case reports. The introduction in clinical practice of newly approved drugs for acute myeloid leukemia will pave the way for new treatment approaches for this rare disease. This report will present a review of acute myeloid leukemia with extramedullary manifestations responding to FLT3-directed target therapy.

**Abstract:**

FMS-like tyrosine kinase 3 (FLT3) is a receptor tyrosine kinase family member. Mutations in *FLT3*, as well known, represent the most common genomic alteration in acute myeloid leukemia (AML), identified in approximately one-third of newly diagnosed adult patients. In recent years, this has represented an important therapeutic target. Drugs such as midostaurin, gilteritinib, and sorafenib, either alone in association with conventional chemotherapy, play a pivotal role in AML therapy with the mutated *FLT3* gene. A current challenge lies in treating forms of AML with extramedullary localization. Here, we describe the general features of myeloid sarcoma and the ability of a targeted drug, i.e., gilteritinib, approved for relapsed or refractory disease, to induce remission of these extramedullary leukemic localizations in AML patients with *FLT3* mutation, analyzing how in the literature, there is an important development of cases describing this promising potential for care.

## 1. Introduction

Acute myeloid leukemias (AML) are still an open challenge today. As known, these are diseases involving the bone marrow, with the presence of immature cellular elements. However, these neoplastic cells are capable of homing in tissues other than bone marrow through mechanisms not yet fully explained. Although a more appropriate name for this manifestation would be “extramedullary acute myeloid leukemia tumor”, the condition is commonly known as myeloid sarcoma (MS). This term is strictly considered by World Health Organization for the cases with no concurrent bone marrow (BM) involvement. A more appropriate name for contemporary manifestation in the BM and another site should be referred to as “extramedullary acute myeloid leukemia tumor”. This is a reality that every hematologist must deal with. In a retrospective analysis of 346 AML patients, extramedullary involvement had an incidence of 11% and was found to significantly complicate prognosis and therapeutic strategies [1]. On the other hand, the improvements in diagnostic techniques and the recent findings in molecular biology (including *FLT3* and its inhibitory drugs) promise to revolutionize the therapeutic workup of this patient setting.

## 2. Features of Extramedullary Localization of AML

The cancer cells of acute myeloid leukemia can sometimes aggregate in extramedullary sites. This extramedullary tumor, known as myeloid sarcoma (MS, also known as granulocytic sarcoma or chloroma), is defined by the World Health Organization as a tumor mass consisting of myeloid blasts with or without maturation occurring at an anatomic site other than BM. MS (or, in general, extramedullary acute myeloid leukemia tumor, e-AML) is composed of cancer cells in various stages of maturation: myeloblasts, monoblasts, and, rarely, megakaryocytic or erythroid precursors [2]. The pathogenesis of e-AML and MS is unknown. It has been associated with the expression of specific cell adhesion molecules, chemokine receptor/ligand interaction, and aberrant FAS-MAPK/ERK signaling. In particular, adhesion molecules, such as CD56 (neural cell adhesion molecule), are more frequently expressed in leukemia cells. CD56 can promote the adhesion of cancer cells through homophilic binding to tissues expressing this molecule (adipose/soft tissue, gastrointestinal, brain, testicular, and skeletal muscle) [3,4,5].

In contrast to this hypothesis, however, a large-scale study showed that the occurrence of CD56-positive leukemic blasts was similar in patients with and without e-AML or MS [6]. Another adhesion molecule is CD11b (surface β2-integrin member macrophage-1 antigen) expressed on mononuclear cells. This molecule mediate the interaction between cells and the matrix and could determine the frequency of extramedullary localization in forms of AML with monoblastic/myelomonocytic differentiation [7,8]. e-AML can have different localizations, and symptoms depend on the involved organ [9]. The most affected organs are skin, soft tissues, and lymph nodes, although the testes, bone, peritoneum, and gastrointestinal tract may also be involved [2]. MS is rarely be isolated (with a finding of 2 new cases per million adult subjects) but more frequently diagnosed in individuals with a previous or concomitant history of other hematological malignancies. Among these, the most frequent are AML, myeloproliferative neoplasm (MPN), myelodysplastic syndrome (MDS), MPN/MDS, and in the blast phase of chronic myeloid leukemia (CML) [10].

Diagnosis starts with the patient’s medical history and clinical suspicion, detailed with instrumental examinations. Among them, computed tomography (CT) is the first imaging modality. CT scans display mild to marked homogeneous post-contrast enhancement. In addition, magnetic resonance imaging (MRI) provides images of the central nervous system (CNS). ^18^Fluorodeoxy-glucose positron emission tomography/CT (FDG-PET/CT) is the gold standard imaging modality to evaluate the presence of e-AML and monitor the response to therapy over time [11,12].

For this reason, FDG-PET/CT is among the mandatory exams of the NCCN for the evaluation of extramedullary localizations of AML. However, a biopsy of the lesion is essential to establish a diagnosis with certainty. A condition of severe thrombocytopenia (PLT < 50,000/mmc) can make biopsy difficult for many suspected organs, particularly those involving the CNS. In such situations, the regression of the lesion following therapy for AML establishes diagnosis with reasonable certainty. The evaluation of the histological section typically reveals infiltrating myeloid cells at various stages of maturation, with either monocytic or granulocytic maturation, in the same way as in AML. Diagnosis is confirmed using immunohistochemistry, an antibody panel commonly used in AML. Specifically, staining for myeloperoxidase (MPO-expressed in 66% to 96% of MS) is used to help differentiate lesions from lymphoma [13]. The other commonly tested antigens are CD43, CD68, lysozyme, myeloperoxidase, CD117, CD11c, CD13, and CD33 [10,14,15]. 

Other techniques that can confirm or improve diagnostic capacity are immunohistochemistry, flow cytometry, and fluorescence in situ hybridization (FISH). A percentage of cases ranging from 25% to 47% are initially mislabeled as melanoma, thymoma, Ewing sarcoma, or malignant lymphoproliferative disorders (e.g., Hodgkin lymphoma, histiocytic lymphoma, mucosa-associated lymphoid tissue lymphoma, large-cell lymphoma) or poorly differentiated carcinoma [14,16,17].

e-AML could also be confused with a blastic plasmacytoid dendritic cell neoplasm (BPDCN), a rare and aggressive hematologic malignancy of the BM and blood that can affect organs such as lymph nodes, skin, spleen, and CNS. Skin lesions are characteristic of this disease; therefore, they must be distinguished from e-AML [18].

NCCN guidelines recommend that patients affected by extramedullary disease should be considered for a screening lumbar puncture evaluation, followed by an empiric dose of intrathecal chemotherapy [19].

The European Society for Hematology (ESH) classifies MS into four subgroups [10]: -Extramedullary involvement with concurrent newly diagnosed AML;-Extramedullary relapse of AML, including in the post bone marrow transplant setting;-Blast phase/transformation of a myeloproliferative neoplasm or chronic myelomonocytic leukemia;-Isolated MS in association with a normal bone marrow biopsy and blood film, as well as the absence of any history of myeloid neoplasia.

Extramedullary leukemic involvement associated with concurrent AML or at relapse occurs more frequently in monoblastic leukemia with translocations involving 11q23, acute myelomonocytic leukemia with eosinophilia, inv(16) (p13;q22) or t(16;16) (p13;q22) [20,21], and in AML with maturation and t(8;21) (q22;q22). Significantly, these cases of MS, among the first described years ago, involving genital organs and frequently associated with AML, formerly known as FAB M2 [22], are characterized by an excellent response to systemic chemotherapy [23].

Several factors influence the choice of best therapy for e-AML. The choice depends on the involved organ, the synchronous presence of medullary involvement of the AML, and any previous lines of treatment for the underlying disease (prior allo-HSCT); furthermore, as already discussed, the choice of therapy is contingent upon the presence of targetable abnormalities (e.g., *FLT3* and *IDH1/2* mutations). For localized forms of the disease, one therapeutic possibility to relieve symptoms may be represented by surgical decompression. MS appears to be sensitive to ionizing radiation, so involved-field radiotherapy (RT) should be considered for all patients with isolated MS refractory to systemic therapy [24]. However, no improvement in five-year survival was found in a retrospective study analyzing patients with extramedullary disease localization who were treated between 1994 and 2006 with systemic therapy either associated or not with RT [25]. For the above reasons, intensive systemic chemotherapy represents the gold standard of e-AML treatment in eligible patients, whether or not associated with bone marrow localization of AML [19]. These patients are treated with classic anthracycline or cytarabine-based therapeutic regimens. In an analysis by Tsimberidou et al., complete remission was achieved by 65% of patients with e-AML, whereas 5% reached a partial remission, with a median overall survival of 20 months (range 1–75) [26].

On the other hand, for patients not eligible for intensive care, hypomethylating agents, e.g., azacitidine (HMA) and deoxyribonucleic acid methyltransferase inhibitor (*DNMT*i) decitabine, appear to have been able to induce complete remission after 2–4 cycles of therapy in few selected clinical cases [27]. Among target therapies, limited data support the use of sorafenib (an FLT3 inhibitor) [28] or drugs such as enasidenib or ivosidenib in the context of the form with *IDH2* or *IDH1* mutations, occurring in 30% and 15% of cases, respectively [29,30].

Further target therapies concern B-cell lymphoma 2 (BCL-2) inhibitors, i.e., venetoclax. In this context, there is no solid scientific evidence yet. Sporadic case reports include e-AML patients treated with venetoclax both in the course of disease refractory to the first line of therapy and in the extramedullary localization of AML arising after HSCT [31,32]. Furthermore, it is interesting to note that there are reports of potential penetration by venetoclax through the cerebrospinal membrane being helpful in treating AML involvement of the CNS [33]. These results certainly require more consistent confirmatory data. 

Regarding the immunotherapeutic approach, success is poor. A proportion of 10% of leukemic cells of the MS overexpress PD-1 [34,35], and immune checkpoints that generally have the task of controlling the degree of inflammation and preserving damage induced by T lymphocytes on healthy tissues and in neoplastic conditions represent an escape from the tumor against the organism’s response to the disease. However, this feature’s clinical and therapeutic significance remains a question to be further investigated [36]. Once complete remission is achieved, consolidation with HSCT is the best therapeutic strategy. The rate of post-allo-HSCT 5-year OS for patients with MS ranges from 47 to 53% [37], comparable to the OS of patients with AML undergoing HSCT (25–40%) [38]. HSCT or donor lymphocyte infusion (DLI) is effective against eAML, probably causing the graft-versus-leukemia (GvL) effect guaranteed by the engrafted immune system (graft-versus-leukemia effect following hematopoietic stem cell transplantation for leukemia). In the case of disease relapse, even after a first bone marrow transplant, a second allo-HSCT can be contemplated for younger patients. However, prospective studies with cytogenetic and molecular data analysis from patients affected by eAML are needed to guarantee an appropriate risk stratification and create a guide for upfront therapy and, eventually, consolidative strategies [39].

## 3. The Role of the *FLT3* Gene in AML and e-AML

Since its discovery by two independent groups in 1991 [40,41], the scientific community has realized that FMS-like tyrosine kinase 3 (*FLT-3*) could play a pivotal role in the study and knowledge of human neoplastic pathologies. The *FLT3* gene, belonging to the class III receptor tyrosine kinase family, encodes a human protein of 993 amino acids and is expressed as a receptor in the placenta, gonads, and brain. *FLT3* receptor activation by its ligand (FL) leads to rapid receptor autophosphorylation and induces the activation of several intermediate signal-transduction mediators, determining cell proliferation and expansion [42,43,44,45,46].

These data demonstrate an essential role for *FLT3* in developing multipotent stem and B cells.

Furthermore, *FLT3* has played a cardinal role in the study and management of AML since 1996, when Yokota et al., using reverse transcriptase-polymerase chain reaction (RT-PCR), found, in 5 AML patient samples (17%), a different transcript with a primer combination that could amplify the transmembrane (TM) domain through the juxtamembrane (JM) domain (*FLT3*-ITD mutation). These repeat sequences disrupted the JM domain’s autoinhibitory activity, resulting in constitutive tyrosine kinase activation [47]. Subsequently, a further mutation was identified involving the activation loop of *FLT3*, a component of the tyrosine kinase domain (*FLT3*-TKD mutation) [48,49,50].

Larger-scale evaluations have shown that *FLT3* mutations are found in approximately 30% of de novo diagnosed AML cases and are divided between ITDs (about 25%) and point mutations in the TKD (7–10%) [51].

Over the years, studies have highlighted the crucial role of these mutations in the context of newly diagnosed AML, as well as their ability to actively characterize the prognosis and, above all, their potential as a therapeutic target. For the *FLT3-TKD* mutation, the prognostic impact is not yet well defined. Despite being based on small samples of patients, the literature data do not confirm a relevant correlation between the presence of this mutation and clinical outcome [45]. On the other hand, the situation is the opposite for *FLT3*-ITD. The higher frequency of incidence of this mutation has allowed us to develop reliable prognosis prospects and make the most appropriate therapeutic decisions. Risk stratification guidelines for AML have been drawn up by several organizations, including the World Health Organization (WHO), the National Comprehensive Cancer Network (NCCN), and the European Leukemia Network (ELN) [19,52], specifying that the presence of *FLT3*-ITD is a genetic alteration that identifies a well-defined subgroup of patients. The NCCN guidelines classify patients with these mutations as having poor prognoses [19]. The same is true for the ELN guidelines, in which the *FLT3* mutation plays a prominent role, specifically as an adverse prognostic indicator of disease. The concept of allelic ratio (AR) represents the number of ITD-mutated alleles compared with the number of the wild-type alleles and therefore is not only influenced by the amount of leukemia versus normal cells in the tested sample but also by the percentages of cells with 0, 1, or 2 mutated alleles. Numerous studies have concluded that a high ratio (AR > 0.5) is associated with worse prognoses and patient outcomes. This concept made it possible to stratify patients into three risk classes (favorable, intermediate, or adverse) depending on the presence or absence of the nucleophosmin member 1 (*NPM1*) gene mutation and to favorably mediate the prognostically adverse effect of *FLT3*-ITD [53]. In contrast, Sakaguchi’s group analyzed 147 patients with *FLT3*-ITD gene mutation-positive AML, stratifying them, according to ELN indications, into high and low AR, depending on the presence or absence of the *NPM1* mutation. They found that *FLT3*-ITD*^low^* AR was not associated with favorable outcomes (overall survival (OS), 41.3%). Furthermore, only patients who underwent allogeneic stem cell transplantation (allo-HSCT) in the first complete remission (CR1) had a more favorable outcome than those who did not (relapse-free survival (RFS) *p* = 0.013; OS *p* = 0.003). It was also underlined by multivariate analysis that allo-HSCT in CR1 is the only prognostic factor capable of giving a better OS and progression-free survival (PFS), demonstrating that prognosis was unfavorable in *NPM1*-mutated AML with *FLT3*-ITD*^low^* AR when allo-HSCT was not carried out in CR1 [54]. Regarding the setting of refractory/relapsed AML patients, there is a risk of onset of leukemic clones with multiple adverse-risk genetic mutations (including *FLT3*-ITD) or the presence, at relapse, of an allelic burden higher than that at diagnosis, capable of negatively influencing the prognosis, unlike the diagnosis, where there is a greater probability of facing a polyclonal disease [55,56,57,58,59,60]. In this context, *FLT3*-ITD mutations are newly detected at relapse more often than *FLT3*-TKD mutations (8% and 2%, respectively) [56], with a prognosis worse than that in patients maintain the wild-type form of the gene [61]. 

In e-AML, the study of molecular markers is crucial for categorizing the disease and guiding clinicians to the best therapeutic choice for patients. Ali Ansari-Lari, M. et al. analyzed 24 e-AML specimens from 20 patients in a study protocol. Clinical information was available for 15 out of 20 patients. e-AML was diagnosed in the setting of AML in nine cases (60%), CML in three cases (20%), and precursor B (Pre-B) ALL in one patient (6.7%), with appearance ranging between 3 months and 21 years after a leukemia diagnosis. In three instances, the e-AML was diagnosed concurrently with leukemia (two AML, one CML). However, six patients had no leukemia bone marrow involvement (it should be emphasized that three cases did not have a concurrent bone marrow aspirate or biopsy). From a molecular point of view, no *FLT3 D835* mutations were identified in the 20 cases examined, whereas *FLT3*-ITD mutations were identified in three of the 20 cases (15%). Contrary to what happens in *FLT3*-mutated AML, their data were too few to identify a reliable prognostic significance [62]. Mutation in the *FLT3*-ITD gene would favor the infiltration of leukemic cells into the visceral organs, simultaneously reducing the BM homing of leukemic cells by deregulating CXCR4 signaling [63]. In the context of extramedullary localizations, as already seen, there are several molecules associated with more significant infiltration of leukemic cells in other organs. As described, CD56 plays an essential role as an adhesion molecule. It is responsible for homing of these cells in several tissues and is highly expressed in the breast, testicular tissue, ovary, and gut. Furthermore, numerous studies have shown that miRNAs play a crucial role in hematopoiesis and hematological malignancies. In this context, FLT3 can modulate the expression of different miRNAs, with both downregulation (miR-451 and miR-144) and upregulation (miR-155, miR-10a, and miR-10b) mechanisms. The expression of these non-coding endogenous small molecules appears to play a role in these conditions, with mechanisms yet to be explored [64].

This evidence explains how *FLT3* mutations occur in a significant subgroup of patients with e-AML. However, as already discussed, if *FLT3*-ITD mutations appear to be associated with increased relapse risk, adverse disease-free survival, and overall survival, drawing a definitive prevalence is not possible due to the limited sample size of analyzed patients. 

As already mentioned, *FLT3*-ITD mutation plays a leading role because, in recent years, the so-called tyrosine kinase inhibitors (TKIs) have been developed to the point that they can bind to this therapeutic target and inhibit its action. Among these, midostaurin has been approved since 2017 in association with chemotherapy with cytarabine and daunorubicin (“7 + 3” scheme) for induction of remission in patients with newly diagnosed AML and in combination with high doses of cytarabine as consolidation therapy. A pivotal drug study (RATIFY) demonstrated improved event-free survival (hazard ratio = 0.78; *p* = 0.002) and OS (hazard ratio = 0.78; *p* = 0.009) compared with 7 + 3 chemotherapy alone in all *FLT3* mutation subtypes (e.g., TKD, ITD^low^*,* and ITD^high^ AR), albeit with different results observed among the single analyzed groups [65]. Other I generation TKIs include lestaurtinib, sunitinib, and sorafenib. Furthermore, several next-generation TKIs have appeared in the therapeutic landscape in the last few years or are still under study. These include drugs such as crenolanib, quizartinib, FLX925, ponatinib, and, especially, gilteritinib. Gilteritinib is a molecule whose efficacy was evaluated in a randomized, open-label phase 3 clinical trial conducted in adult patients with relapsed or refractory AML with *FLT3* gene mutation (ADMIRAL study). In this setting, 371 patients were randomly treated, receive gilteritinib or any other salvage chemotherapy (low-dose cytarabine, azacitidine, combination of chemotherapeutic agents according to the MEC or FLAG-Ida scheme). Patients randomized to the gilteritinib arm had significantly longer survival than those in the chemotherapy arm, with a median OS of 9.3 months versus 5.6 months, respectively. This was also confirmed by a higher rate of complete remission with complete (21.1% for gilteritinib arm versus 10.5% for chemotherapy arm) or partial (13% for gilteritinib arm versus 4.8% for chemotherapy arm) hematological recovery [66]. Concerning MS or eAML in general, as previously mentioned, the presence of these types of mutations and the increasing use of FLT3 inhibitory molecules suggest their potential as therapeutic targets to be carefully analyzed [67].

For all these reasons, a rapid *FLT3*-ITD diagnostic assay can identify cases of patients suffering from AML with a poor prognosis and guide clinicians’ therapeutic choices, providing patients the chance to improve their survival thanks to the use of *FLT3*-ITD targeted therapies [52,53,68].

## 4. Myeloid Sarcoma and Gilteritinib Data from Real Life

In the context of *FLT3*-mutated AML with extramedullary involvement, gilteritinib is the inhibitor drug with the best response. In this work, we focused on collecting the few clinical cases reported in the literature demonstrating the therapeutic success of gilteritinib in MS (Table 1).

In the first case [69], the authors describe the history of a 56-year-old patient affected by *FLT3*-ITD-mutated AML and undergoing bone marrow transplantation, with complete remission and complete donor chimerism evaluated on days +27 and +96 after transplant. On day +180, several subcutaneous lesions appeared in the follow-up of extensive chronic GVHD. Localization of myeloid sarcoma was diagnosed upon histological examination. *FLT3*-ITD mutation was still present. Maintaining his bone marrow complete donor chimerism was considered a case of isolated extramedullary relapse. Despite treatment with etoposide and ranimustine, tumor progression continued. Second-line therapy with gilteritinib was started, finally achieving remarkable regression of the tumors and complete disappearance of skin disease.

In another case [70], a 38-year-old male patient was diagnosed with AML with myelodysplasia-related changes and *FLT3*-ITD mutation. He was subjected to therapy with cytarabine and idarubicin but without obtaining a response and maintaining profound pancytopenia; hence, a hematopoietic stem cell transplant was performed directly. The post-transplant clinical course was complicated by an early increase in Wilms tumor gene-1 (*WT-1*) expression and GVHD onset. He was treated with donor lymphocyte infusion and immunosuppressive therapy. On day + 400, a right supraclavicular mass was observed simultaneously with the appearance of AML blasts in bone marrow (6.9%). Histological examination of the tumor lesion identified many cells with atypical nuclei and scant cytoplasm that were immunohistologically positive for CD68, MPO, and lysozyme; slightly positive for CD117; and negative for CD34 and terminal deoxynucleotidyl transferase. Extramedullary and medullary cells harbored *FLT3*-ITD mutation. Therapy with 120 mg/day of gilteritinib was abruptly begun, achieving a bone marrow and MS complete response to treatment, allowing the patient to undergo a second allo-HSCT, achieving a new complete molecular remission, maintained even 150 days after the cell infusion.

Another patient [71] with *NPM1* and *FLT3*-ITD-mutated therapy-related AML was treated by induction and consolidation therapy with CPX-351, obtaining a complete response with positive MRD. Nevertheless, he suddenly began to report burning leg pain and weakening of the right hand and leg muscles, associated with the absence of osteotendinous leg reflexes. A meningeal relapse of AML was confirmed upon examination of cerebrospinal fluid and magnetic resonance imaging, demonstrating two meningeal implants of myeloid sarcoma. The disease progressed despite a brief second response to treatment with medicated lumbar puncture and FLA-Ida chemotherapy. For this reason, it was treated with gilteritinib. After 3 months of therapy, although only a partial response to the bone marrow evaluation of the condition was obtained, the meningeal localization of the disease progressively decreased until it completely disappeared, demonstrating for the first time the potential biological effect of these drugs within the central nervous system.

Localizations of e-AML in infrequent organs are possible. The case of a 60-year-old patient with *FLT3*-ITD mutated AML involving the skin and treated with induction therapy based on cytarabine and idarubicin is described [72], with the achievement of complete remission. New hyperleukocytosis (18,900/mmc) was found during consolidation therapy, confirming a new disease relapse. At the same time, the patient began to experience skin-onset photophobia and loss of visual acuity. Right ocular examination revealed a yellow-white infiltrative mass in the right eye involving the temporal iris, ciliary body, and choroid. Fine-needle aspiration biopsy was performed, confirming the extramedullary localization of the leukemic process and establishing the diagnosis of myeloid sarcoma. Given the *FLT3*-ITD mutation, the patient began treatment with oral gilteritinib, with rapid regression of the tumor, complete disappearance of the iris component, and a significant reduction in the size of the ciliochoroidal tumor accompanied by an improvement in visual acuity. Despite the excellent response of the extramedullary part of the AML, the patient died due to the progression of the medullary AML.

Lastly, a 47-year-old female [73] was diagnosed with *NPM1*-mutated and *FLT3* wild-type AML and treated with “7 + 3” cytarabine/anthracycline induction and four cycles of high-dose cytarabine consolidation, obtaining complete remission (CR). After a first relapse of the disease, she was enrolled in a clinical trial and underwent HSCT. Three months post-transplant, she had disease relapse, with a new appearance of *FLT3*-ITD mutation at a low mutant-to-wild type allelic ratio of 0.27. A new CR was achieved with two cycles of sorafenib (a first-generation TKI inhibitor) and azacitidine. Sorafenib was continued on maintenance therapy. Sixteen months later, she developed a new vaginal mass and bilateral breast masses, with ^18^fluorodeoxy-glucose avidity on positron emission tomography/computed tomography (PET/CT). Biopsy of the lesion confirmed MS. Marrow evaluation was normal. NGS testing of MS revealed *FLT3*-ITD mutation. She began treatment with 120 mg of gilteritinib daily, and after 1 month of therapy, a repeat PET/CT showed a fourth CR, and the patient was able to undergo a second life-saving HSCT.

Despite the increase in cases of extramedullary localizations of leukemic cells found at various times during AML and described in the literature, there are still too many limitations and uncertainties to conclude with certainty. Specifically, there is a lack of knowledge comparing the various FLT3 inhibitors, where only gilteritinib manages to have more critical data to support its use, with its efficacy probably due to its mechanisms of action. Another explanation could lie in the potential of the drug to penetrate into the extramedullary environment, whereas in the context of the bone marrow, the medullary niches in these hematological malignancies could guarantee an environment of survival and escape of cancer cells from standard drugs. For these reasons, more data and evidence from real-life settings are needed. 

## 5. Discussion

Historically, the response to induction therapies in bone marrow or extramedullary forms of AML has been similar between patients with and without *FLT3* mutations. However, as already seen, the shorter duration of responses and the poor results obtained at the time of relapse with salvage therapies have led to a significant reduction in overall survival for patients with *FLT3*-ITD mutations [74]. Although the advent of *FLT3* inhibitors has dramatically changed the therapeutic prospects and the possibility of survival of this subset of patients, both in association with traditional chemotherapy and in monotherapy [51], the short-lived response following the onset of resistance mechanisms remains an open challenge [75]. Intrinsic mechanisms (internal within AML cells) can be divided into primary (occurring before treatment) or secondary (induced by FLT3 inhibitor therapy) mechanisms. Among the most important is the assumption that an AML in relapse has a high risk of selecting a mutated *FLT3* clone with a higher AR than the diagnosis, when there is a greater probability of facing a polyclonal disease. For this reason, newly diagnosed *FLT3*-mutated AML might be less likely to respond clinically to highly selective *FLT3* inhibition [60]. Secondly, upregulation of the antiapoptotic proteins Bcl-xL and Mcl-1 could be triggered by *FLT3*-ITD*627E* mutation [76,77].

The same primary intrinsic mechanism could involve the upregulation of the Bcl-2 protein, the expression levels of which were not reduced after dephosphorylation of FLT3 and its downstream target, STAT5, in patient samples with *FLT3*-ITD [78]. Genomic instability activation of AXL or SYK is another well-known intrinsic resistance mechanism secondary to treatment with FLT3 inhibitors. Specifically, activated AXL is reported to be responsible for resistance to FLT3 inhibitors, such as quizartinib and midostaurin, in *FLT3*-ITD mutated AML cells. Gilteritinib could also exert its mechanism of action, inhibiting this target [79]. On the other hand, the medullary microenvironment plays a pivotal role among the extrinsic resistance mechanisms. The increase in the production of FLT3 ligand after chemotherapy is a process that stimulates FLT3-expressing cells, significantly decreasing susceptibility to inhibitory drugs [80,81].

The same phenomenon happens with the secretion of fibroblast growth factor 2 (FGF2) [82]. Extensive studies and in-depth research that have been carried out mainly concern the concept of an inflammatory niche, a space within the bone marrow where cancer cells can survive and evade the toxic effects of standard therapies. Numerous cell types are involved in maintaining this niche [83]. Significantly, the eosinophils located nearby are remarkably proliferating, representing one of the major players in the survival of the inflammatory niche [84]. The localization within the bone marrow of the cells involved in the survival of the inflammatory survival niche is mediated by chemokine (CXC motif) ligand 12 (CXCL12) [85]. Resistance to FLT3 inhibitors is greater following the activation of CXCL12–CXCR4–mediated homing. This may be due to an action of *FLT3*-ITD mutation, which leads to an increase in chemotaxis by blocking the downregulation of Rho-associated kinase via the CXCL12–CXCR4 signaling axis [86], both through the action of the PIM*-1* protein (a member of PIM family of serine/threonine kinases) and the phosphorylation of CXCR4, enabling its expression [87]. The onset of new resistant mutations among other patient-related resistance mechanisms may occur. Drugs such as quizartinib and sorafenib, when used against mutated forms of *FLT3*-ITD mutated AML, can induce TKD mutations, against which they are ineffective [88]. Among the FLT3 inhibitors, our attention is focused on the role of gilteritinib as a therapy of choice in *FLT3*-mutated AML in relapse. In this context, an explanation of the potential of gilteritinib to act in the extramedullary localizations of the disease can be hypothesized by evaluating the estimates of the volume of distribution of this drug, both central and peripheral. They were found to be 1092 L and 1100 L, respectively. This finding indicates a wide distribution of extraplasmic gilteritinib, which could mean widespread tissue diffusion.

For this reason, gilteritinib could have greater efficacy in MS compared to the ability of blast cell clearance in bone marrow, where the inflammatory survival niche allows for avoidance of the response. The last mechanism of the ineffectiveness of the drug could be related to its mode of transport. The binding of gilteritinib to plasma proteins in vivo is approximately 90% in humans, binding mainly to albumin. For this reason, albumin could have a reduced penetrance within bone marrow or, particularly in debilitated patients, albumin deficiency could be an interesting clinical indicator/predictor of drug response. Further studies are needed to confirm these hypotheses, especially in the field of MS, where the complexity of diagnosis, clinical characteristics of the patient, and the rarity of the disease still require more significant support.

## 6. Conclusions

The gold-standard therapy for extramedullary acute myeloid leukemia and MS is still an open challenge, as there are no clear indications as to which treatment is suitable for each patient. Indeed, the advent of molecular biology studies and progress in the field of target therapy have opened up new landscapes. In this context, gilteritinib could represent a valid therapeutic option for patients with extramedullary involvement associated with FLT3-mutated AML at relapse, especially for its efficacy on the extramedullary component. New preclinical and clinical data continue to be rapidly generated, intending to guarantee the best successful targeted therapy for this uncommon type of hematological disease.

## Figures and Tables

**Table 1 cancers-14-02186-t001:** Panel of clinical cases reported in the literature treated with gilteritinib, as described in the text. AML = acute myeloid leukemia; MRC-AML = myelodysplasia-related-change AML; t-AML = therapy-related AML; wt = wild type; HSCT = hematopoietic stem cell transplant; DLI = donor lymphocyte infusion; HDAC = high-dose Ara-C; CR = complete remission; PR = partial remission.

Case No.	Patient Features at AML Diagnosis	Previous Treatments for AML	MS Localization	MS Onset	MS Therapies	Outcome
1	56-year-old patient affected by *FLT3*-ITD-mutated AML	Cytarabine/anthracycline induction scheme and HSCT	Subcutaneous all over the soma	On day +180 from HSCT	Etoposide and ranimustine, Gilteritinib	Complete disappearance of skin disease
2	38-year-old patient with diagnosis of *FLT3*-ITD-mutated MRC-AML	Cytarabine/anthracycline induction scheme and HSCT, DLI	Supraclavicular mass	On day +400 from HSCT	Gilteritinib	New CR in the bone marrow and extramedullary involvement, second HSCT administration
3	Patient affected by *NPM1* and *FLT3*-ITD-mutated t-AML	CPX-351 for induction and consolidation therapy	Cerebrospinal fluid and meningeal	During follow-up after the end of consolidation therapy	FLA-Ida and medicated LP, Gilteritinib	Complete regression of meningeal relapse, PR in bone marrow re-evaluation
4	60-year-old patient affected by *FLT3*-ITD-mutated AML involving the skin	Cytarabine/anthracycline induction scheme	Irido-cilio-choroidal	During consolidation therapy	Gilteritinib	Complete regression of extramedullary involvement and improvement of visual acuity, but a contemporary progression of AML, leading to patient death
5	47-year-old patient affected by *NPM1*-mut and *FLT3* wt AML	Cytarabine/anthracycline induction scheme and four consolidation cycles of HDAC, clinical trial and HSCT, sorafenib, and azacytidine caused *FLT3*+ relapse	Vaginal and bilateral breast masses	In third overall *FLT3*+ AML relapse, second after HSCT	Gilteritinib	Fourth CR and second HSCT administration

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
