# Peer review of "Target Therapy for Extramedullary Relapse of FLT3-ITD Acute Myeloid Leukemia: Emerging Data from the Field"

_cancers, 2022, doi:10.3390/cancers14092186_

Round 1
Reviewer 1 Report
The authors review the clinical feature of “Myeloid Sarcoma”, FLT3 mutations in AML, and summarized 5 cases with extra medullary relapse of FLT3-ITD+ AML, which have been reported in literature. However, there are some critical problems with this paper.
The WHO definition of "Myeloid Sarcoma (MS)" is mentioned in Sections 1 and 2. Since MS refers to a disease entity that causes myeloid blast mass formation without bone marrow infiltration, here “Extramedullary involvement in AML” should be reviewed as a pathological condition.
Second, Section 3 contains a review of FLT3 in AML, but should focus on its implications for organ infiltration and tumorigenicity.
Third, Section 4 (although it is Section 3 in the original text) 5 case reports are summarized. Readers may be impressed with the effectiveness of gilteritinib, but it should also be mentioned that the five case reports alone have various limitations. The organ pharmacokinetics of gilteritinib should also be also discussed compared with other FLT3-inhibitors.
Finally, all "MS" in Table 1 should be changed to “Extramedullary involvement”.
Author Response
Response to Reviewer 1 Comments
Point 1: The WHO definition of "Myeloid Sarcoma (MS)" is mentioned in Sections 1 and 2. Since MS refers to a disease entity that causes myeloid blast mass formation without bone marrow infiltration, here “Extramedullary involvement in AML” should be reviewed as a pathological condition.
Response 1: We thank you for your suggestion; we have reviewed the entire manuscript, specifying when it comes to myeloid sarcoma (unique extramedullary localization) or extramedullary involvement of AML in cases where it is a contemporary manifestation with bone marrow disease, as well as the WHO definition
Point 2: Section 3 contains a review of FLT3 in AML, but should focus on its implications for organ infiltration and tumorigenicity.
Response 2: We have proceeded to specify in a better and more detailed way the discussion on the role of FLT3 in the setting of organ infiltration
Point 3: Section 4 (although it is Section 3 in the original text) 5 case reports are summarized. Readers may be impressed with the effectiveness of gilteritinib, but it should also be mentioned that the five case reports alone have various limitations. The organ pharmacokinetics of gilteritinib should also be also discussed compared with other FLT3-inhibitors.
Response 3: We have corrected the title of the paragraph and proceeded to specify hypotheses for which gilteritinib could be effective in this disease setting, confirming, however, that further evidence from real life is necessary to establish or not such beliefs.
Point 4: All "MS" in Table 1 should be changed to “Extramedullary involvement”
Response 4: All "MS" in Table 1 have been changed, as suggested

Reviewer 2 Report
The review paper on targeted therapy for extramedullary relapse of FLT3-ITD acute myeloid leukemia with the clinical summary of experiences in treating actual patient with focus on gilteritinib, by Andrea Duminuco et al. is an interesting and valuable addition to the expertise on acute myeloid leukemia treatment. The work is comprehensive, well written and informative with extensive discussion on biological and clinical aspects of the targeted therapy with FLT3 inhibitor(s). The only suggestion to the authors is to include a reference of one of the first reports on presence on t(8;21) rearrangement in MS, the fact that are authors correctly mentioning but with no specific reference (textual line 122-124). Please see Thalhammer F, Gisslinger H, Chott A, et al. Granulocytic sarcoma of the prostate as the first manifestation of a late relapse of acute myelogenous leukemia. Ann Hematol. 1994 Feb;68(2):97-9. doi: 10.1007/BF01715141.
Author Response
Response to Reviewer 2 Comments
Point 1: The only suggestion to the authors is to include a reference of one of the first reports on presence on t(8;21) rearrangement in MS, the fact that are authors correctly mentioning but with no specific reference (textual line 122-124). Please see Thalhammer F, Gisslinger H, Chott A, et al. Granulocytic sarcoma of the prostate as the first manifestation of a late relapse of acute myelogenous leukemia. Ann Hematol. 1994 Feb;68(2):97-9. doi: 10.1007/BF01715141.
Response 1: We thank you for your revision; we have reviewed the entire manuscript and modified the paragraph to include the suggested reference

Round 2
Reviewer 1 Report
I think revised manuscript is almost acceptable.
Reference #4, 46 and 48 should be fully described.